# Data-Centric Unlearning: Optimizing Labels and Retain Data via Learning Dynamics

## Abstract

Machine unlearning mitigates adverse effects from erroneous, outdated, or private training data. Although unlearning algorithms have advanced for classifiers and LLMs, the critical role of unlearning training data quality remains largely unexplored. This work addresses this fundamental gap by systematically investigating how to construct effective unlearning training sets, focusing on optimal label assignment for samples and strategic selection for the retain set. We leverage learning dynamics theory to analyze the impact of training data on unlearning performance. Precisely, we derive: (1) an optimal label assignment scheme for both unlearning and retain samples, and (2) the principle that neighborhood and boundary samples are most beneficial for inclusion in the retain set. We translate these theoretical insights into data optimization algorithms tailored for both classifiers and LLMs unlearning. Extensive experiments across classifier and LLMs unlearning tasks demonstrate that our data optimization strategies significantly enhance the performance of existing SOTA unlearning algorithms. Our work establishes data optimization as a crucial pillar for effective machine unlearning.

## 1 Introduction

Machine unlearning has become an essential capability for adaptive AI systems, enabling the removal of incorrect, outdated, or private knowledge from trained models to meet key needs in privacy, regulation, and accuracy (Sekhari et al., 2021; Nguyen et al., 2025). Fundamentally, this process operates through parameter updates that strategically attenuate a model's original responses (whether labels or generated outputs) to designated forget samples while preserving generalization performance on unrelated data. This dual objective necessitates two core datasets: the forget set containing target knowledge for removal, and the retain set preserving model capability. For classifier, unlearning is typically executed via supervised fine-tuning (SFT) across both core sets (Cha et al., 2024; Zhou et al., 2025). For LLMs, methodologies extend beyond SFT to include parameter-efficient techniques (e.g., LoRA), in-context learning adjustments, prefiltering, and prompt engineering.

Despite much progress in unlearning for classifiers and LLMs, a key issue is missed: how to improve the training data in unlearning. Current methods have clear flaws. For classifiers, labels of forget samples are often randomly reassigned. For LLMs, outputs are usually replaced with replies like "I don't know" or random text (Gao et al., 2025). Earlier studies show that such schemes lead to shallow unlearning (where old knowledge remains) or new issues (Liu et al., 2025; Yuan et al., 2025). Additionally, most work either assumes retain sets are given or tries to make up for missing data. We believe good unlearning depends on well-built forget and retain sets—just like good data leads to better performance in normal machine learning. Specifically, two data problems need solving: (1) how to assign better labels to both forget and retain samples, and (2) how to wisely choose retain samples that keep the model useful—a gap our work aims to fill.

This work builds on the learning dynamics theory proposed by Ren & Sutherland (2025). Originally developed for analyzing fine-tuning in LLMs, it also applies to conventional classifiers. It explains how each training sample affects the model's predictions on other data during learning. We use this framework to tackle the main goal of machine unlearning: altering the model's responses to forget samples while preserving its behavior on retain samples. Using tools from this theory, we

develop principles for improving label assignment and selecting retrain data. Importantly, our label optimization lets users set target unlearning strength per forget sample. The system then produces optimal labels tailored to that forgetting level. Our analysis shows that an effective retain set should include nearby and decision-boundary samples. From these insights, we design techniques for both classification models and LLMs. For classifiers, we derive exact closed-form label assignments. For LLMs, we develop efficient approximations, bridging theory and practice.

Our theoretical principles also help explain why some recent methods work well—such as those using adversarial examples (Ebrahimpour-Boroojeny et al., 2025), neighbor samples (Maini et al., 2024; Li et al., 2025), or improved label schemes (Yang, 2025). We thoroughly evaluated our approach on both classifiers and LLMs. Extensive experiments show that our method consistently improves existing top techniques across multiple metrics. Ablation and sensitivity studies further confirm the strength of our framework.

## 2 THEORETICAL ANALYSIS WITH LEARNING DYNAMICS

### 2.1 PRELIMINARY FOR LEARNING DYNAMICS

Let $\pi_{\boldsymbol{\theta}}$ represent a model. $\mathbf{x}_n$ and $\mathbf{y}_n$ denote a new training sample and its label (or response in LLMs), respectively. Let $\mathbf{x}_o$ be another sample and its model output (or response in LLMs) $\mathbf{y}_o = \pi_{\boldsymbol{\theta}^t}(\cdot \mid \mathbf{x}_o)$. If $\mathbf{x}_n$ is used for training, then the one-step model update is $\Delta\boldsymbol{\theta} \triangleq \boldsymbol{\theta}^{t+1} - \boldsymbol{\theta}^t = -\eta \cdot \nabla\mathcal{L}\left(f_{\boldsymbol{\theta}}(\mathbf{x}_n), \mathbf{y}_n\right)$. Ren & Sutherland (2025) then defined the learning dynamics ($\mathcal{LD}$) on $\mathbf{x}_o$ is

$$\mathcal{LD}(\mathbf{x}_o, \mathbf{x}_n, \mathbf{y}_n; \pi_{\boldsymbol{\theta}^t}) \triangleq \log \pi_{\boldsymbol{\theta}^{t+1}}(\cdot \mid \mathbf{x}_o) - \log \pi_{\boldsymbol{\theta}^t}(\cdot \mid \mathbf{x}_o), \tag{1}$$

where $\pi(\cdot \mid \mathbf{x}_o)$ represents the output probability vector over all labels (or tokens). For a classification task with $C$ categories, let $\mathbf{z}_n$ be the logit vector for $\mathbf{x}_n$. Eq. (1) can be decomposed into

$$\mathcal{LD}(\mathbf{x}_o, \mathbf{x}_n, \mathbf{y}_n; \pi_{\boldsymbol{\theta}^t}) = -\eta\mathcal{A}^t(\mathbf{x}_o)\mathcal{K}^t(\mathbf{x}_o, \mathbf{x}_n)\mathcal{G}^t(\mathbf{x}_n, \mathbf{y}_n) + \mathcal{O}(\eta^2\|\nabla_{\boldsymbol{\theta}}\mathbf{z}_n\|_{\mathsf{op}}^2), \tag{2}$$

where the three key terms $\mathcal{A}^t(\cdot \mid \mathbf{x}_o) = \nabla_{\mathbf{z}}\log\pi_{\boldsymbol{\theta}^t}(\mathbf{x}_o) = I - \mathbf{1}\pi_{\boldsymbol{\theta}^t}^\top(\cdot \mid \mathbf{x}_o)$, $\mathcal{K}^t(\mathbf{x}_o, \mathbf{x}_n) = (\nabla_{\boldsymbol{\theta}}\mathbf{z}(\mathbf{x}_o)|_{\boldsymbol{\theta}^t})(\nabla_{\boldsymbol{\theta}}\mathbf{z}(\mathbf{x}_n)|_{\boldsymbol{\theta}^t})^\top$, and $\mathcal{G}^t(\mathbf{x}_n, \mathbf{y}_n) = \nabla_{\mathbf{z}}\mathcal{L}(\mathbf{x}_n, \mathbf{y}_n)|_{\mathbf{z}_n^t}$. The specific forms of $\mathcal{A}$, $\mathcal{K}$, and $\mathcal{G}$ depend on the learning tasks and are provided in (Ren & Sutherland, 2025). Although an LLM is a highly complex model, it can be viewed as a multi-label classifier on $V$ vocabularies. Consequently, the standard multi-class classification scenario is adopted for further analysis, and conclusions derived from it are generally valid for LLMs. In classification, when the cross-entropy loss is employed, $\mathcal{G}_{\mathrm{CE}}^t(\mathbf{x}_n, \mathbf{y}_n) = \pi_{\boldsymbol{\theta}^t}(\cdot \mid \mathbf{x}_n) - \mathbf{y}_n$, which is a $C$-dimensional vector.

Although $\mathcal{LD}$ defined in Eq. (1) is a one-step update, Ren & Sutherland (2025) assumed that during the training, the relative influence of learning $\mathbf{x}_n$ on $\mathbf{x}_o$ is relatively stable. As their theory is well verified in their study, this assumption is also followed in this study. The second term in the right side of Eq. (2) can be omitted in the following analysis.

### 2.2 ANALYSIS FOR MACHINE UNLEARNING

We first define a slight variation of $\mathcal{LD}$, namely relative learning dynamics ($\mathcal{RLD}$), to quantify the relative change in the output distribution induced by a training step with a training sample $(\mathbf{x}, \mathbf{y})$:

$$\mathcal{RLD}(\mathbf{x}_o, \mathbf{x}, \mathbf{y}; \pi_{\boldsymbol{\theta}^t}) \triangleq \frac{\log\pi_{\boldsymbol{\theta}^{t+1}}(\cdot \mid \mathbf{x}_o) - \log\pi_{\boldsymbol{\theta}^t}(\cdot \mid \mathbf{x}_o)}{\|\log\pi_{\boldsymbol{\theta}^t}(\cdot \mid \mathbf{x}_o)\|_2}. \tag{3}$$

This measure better characterizes the intrinsic influence of the training sample on $\mathbf{x}_o$'s prediction, as it normalizes the absolute change by the initial state. For instance, consider two samples $\mathbf{x}_{o1}$ and $\mathbf{x}_{o2}$ with initial predictions $\pi_{\boldsymbol{\theta}^t}(\cdot \mid \mathbf{x}_{o1}) = [0.9, 0.1]^\top$ and $\pi_{\boldsymbol{\theta}^t}(\cdot \mid \mathbf{x}_{o2}) = [0.55, 0.45]^\top$ in a binary classification task. After an update, suppose they become $[0.7, 0.3]^\top$ and $[0.4, 0.5]^\top$, respectively. $\mathcal{LD}$ vector norm (over labels) is larger for $\mathbf{x}_{o1}$ ($\|\mathcal{LD}\| \approx 1.13$) than for $\mathbf{x}_{o2}$ ($\|\mathcal{LD}\| \approx 0.43$), suggesting a greater effect on $\mathbf{x}_{o1}$. However, intuitively, the influence on $\mathbf{x}_{o2}$ is more critical because its predicted label actually changes. Using $\mathcal{RLD}$, we obtain $\|\mathcal{RLD}(\mathbf{x}_{o1})\| \approx 1.03$ and $\|\mathcal{RLD}(\mathbf{x}_{o2})\| \approx 1.49$, which aligns better with intuition by accounting for the relative shift. Fig. 1(a) and (b) show the heatmaps of $\|\mathcal{LD}\|$ and $\|\mathcal{RLD}\|$ after the change of the same classifier (represented by the original

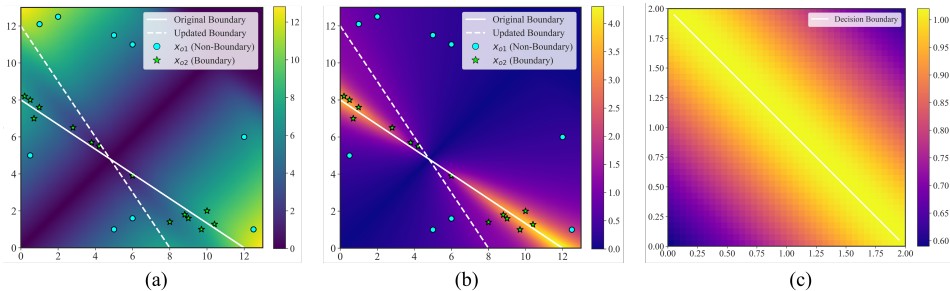

(a)                     (b)                    (c)

Figure 1: (a) The heatmap of $||\mathcal{LD}||$ after a classifier update (decision boundary changes); (b) The heatmap of $||\mathcal{RLD}||$ after the same classifier update with (a); (c) The heatmap of $\frac{||\mathcal{A}^t(\mathbf{x})||_F}{||\log \pi_{\boldsymbol{\theta}^t}(\cdot|\mathbf{x})||_2}$.

boundary and the updated boundary), respectively. In Fig. 1(a), some points far from both classification planes have significantly large $||\mathcal{LD}||$, while some points with changed predicted categories have relatively small $||\mathcal{LD}||$. In Fig. 1(b), regions with large $||\mathcal{RLD}||$ are concentrated near the original boundary and in the areas where predicted categories have changed. These regions are indeed more significantly affected by the classifier's change.

Given a forget set $\mathcal{FS} = \{\mathbf{x}_i, \mathbf{y}_i\}_{i=1}^{N_u}$ and a retain set $\mathcal{RS} = \{\mathbf{x}_j, \mathbf{y}_j\}_{j=1}^{N_r}$, machine unlearning updates the current model with the following learning object:

$$\boldsymbol{\theta}^* = arg\min_{\boldsymbol{\theta}}[\frac{1}{N_u}\sum_i \mathcal{L}_u(\mathbf{x}_i, \mathbf{y}_i; \boldsymbol{\theta}) + \frac{1}{N_r}\sum_j \mathcal{L}_r(\mathbf{x}_j, \mathbf{y}_j; \boldsymbol{\theta})], \quad (4)$$

where $\mathcal{L}_u$ and $\mathcal{L}_r$ are loss functions (identical or different). Eq. (4) characterizes two machine unlearning sub-goals: (1) forgetting model outputs for $\mathcal{FS}$ samples, and (2) minimizing the model's output impact on $\mathcal{RS}$ samples. In practice, each $\mathbf{x}_i$ in $\mathcal{FS}$ is fixed and provided, while its $\mathbf{y}_i$ must be assigned; $\mathcal{RS}$ is also provided, but $\{\mathbf{y}_j\}$ assignment and $\mathbf{x}_j$ retention remain undetermined. In short, $\{\mathbf{y}_i\}$, $\{\mathbf{y}_j\}$, and $\{\mathbf{x}_j\}$ have considerable room for optimization.

Guided by learning dynamics theory, we propose the following optimization insights: $\{\mathbf{y}_i\}$, $\{\mathbf{y}_j\}$ assignment should minimize their $\mathcal{RLD}$ on non-unlearning samples; meanwhile, prioritize $\{\mathbf{x}_j\}$ to select samples most affected by unlearning data. The first is intuitive, while the second aims to select highly impacted samples in the retain set, thereby reducing the model's overall update effect on non-unlearning samples. The two insights can be expressed as the following form:

$$\mathbf{y}^* = arg\min_{\mathbf{y}} ||\mathcal{RLD}(\mathbf{x}_o, \mathbf{x}, \mathbf{y}; \pi_{\boldsymbol{\theta}^t})||, \text{ for } \mathbf{x} \in \mathcal{FS} \text{ or } \mathcal{RS}, \forall \mathbf{x}_o \notin \mathcal{FS}$$

$$\mathbf{x}^* = arg\max_{\mathbf{x} \in \mathcal{RS}} ||\mathcal{RLD}(\mathbf{x}, \mathbf{x}_i, \mathbf{y}_i^*; \pi_{\boldsymbol{\theta}^t})||, \text{ for each } \mathbf{x}_i \in \mathcal{FS} \quad (5)$$

The first equation describes how to obtain the optimal label for both a forget[1] and a retain sample; the second equation describes how to select an optimal retain sample from $\mathcal{RS}$ according to a forget sample. We first derive the optimal label assignment for retain samples. In the multi-class scenario,

$$||\mathcal{RLD}(\mathbf{x}_o, \mathbf{x}, \mathbf{y}; \pi_{\boldsymbol{\theta}^t})||_2 \le \eta ||\mathcal{A}^t(\mathbf{x}_o)||_F ||\mathcal{K}^t(\mathbf{x}_o, \mathbf{x})||_F ||\mathcal{G}_{CE}^t(\mathbf{x}, \mathbf{y})||_2 / ||\log \pi_{\boldsymbol{\theta}^t}(\cdot \mid \mathbf{x}_o)||_2. \quad (6)$$

$\mathbf{y}$ only influences $||\mathcal{G}_{CE}^t(\mathbf{x}, \mathbf{y})||_2$ in (6). Therefore, the first objective of (5) for retain data becomes

$$\mathbf{y}^* \approx arg\min_{\mathbf{y}} ||\mathcal{G}_{CE}^t(\mathbf{x}, \mathbf{y})||_2 = arg\min_{\mathbf{y}} ||\pi_{\boldsymbol{\theta}^t}(\cdot \mid \mathbf{x}) - \mathbf{y}||_2. \quad (7)$$

Obviously, the solution is $\mathbf{y}^* = \pi_{\boldsymbol{\theta}^t}(\cdot \mid \mathbf{x})$. That is, the optimal label of a retain sample is the model output (i.e., Softmax vector in classification) rather than the true label of the sample, which well explains why previous unlearning studies tend to favor the KL divergence loss for retain samples.

Likewise, for the second object in (5), we have

$$\mathbf{x}^* = arg\max_{\mathbf{x} \in \mathcal{RS}} \frac{||\mathcal{A}^t(\mathbf{x})||_F}{||\log \pi_{\boldsymbol{\theta}^t}(\cdot \mid \mathbf{x})||_2} ||\mathcal{K}^t(\mathbf{x}, \mathbf{x}_i)||_F, \text{ for each } \mathbf{x}_i \in \mathcal{FS}. \quad (8)$$

This optimization has excessively high complexity to solve directly. Therefore, we conduct independently analysis for the two items in the right side of (8). First, we explore under what conditions $\frac{||\mathcal{A}^t(\mathbf{x})||_F}{||\log \pi_{\boldsymbol{\theta}^t}(\cdot|\mathbf{x})||_2}$ can attain its maximum. For a linear binary classification problem, we have:

---

[1]When $\mathbf{x} \in \mathcal{FS}$, $\mathbf{y}^*$ should be different from $\pi_{\boldsymbol{\theta}^t}(\cdot|\mathbf{x})$, which will be discussed in Section 3.1.

**Lemma 1** *For a given sample* $\mathbf{x}$*, as it approaches the decision boundary, the ratio* $\frac{||\mathcal{A}^t(\mathbf{x})||_F}{||\log \pi_{\boldsymbol{\theta}^t}(\cdot|\mathbf{x})||_2}$ *increases monotonically and attains its maximum when* $\mathbf{x}$ *lies exactly on the boundary.*

The proof and multi-class extension are presented in Appendix A.1. For the multi-class case, quite similar conclusions can also be obtained. Move over, the more classes the boundary refers to, the larger the value. Fig. 1(c) shows illustrative examples for the binary case.

Second, we analysis that under what conditions the term $||\mathcal{K}^t(\mathbf{x}, \mathbf{x}_i)||_F$ increases. Let $\nabla_{\boldsymbol{\theta}}\mathbf{z}(\mathbf{x})_{|\boldsymbol{\theta}^t} = [\mathbf{g}_1, \cdots, \mathbf{g}_C]$ and $\nabla_{\boldsymbol{\theta}}\mathbf{z}(\mathbf{x}_i)_{|\boldsymbol{\theta}^t} = [\mathbf{g}_{i,1}, \cdots, \mathbf{g}_{i,C}]$. We have $||\mathcal{K}^t(\mathbf{x}, \mathbf{x}_i)||_F = ||\sum_{l=1}^{C} \mathbf{g}_l \mathbf{g}_{i,l}^\top||_F$. Therefore, the value of $||\mathcal{K}^t(\mathbf{x}, \mathbf{x}_i)||_F$ depends on two aspects: 1) the alignment of between the directions of $\mathbf{g}_l$ and $\mathbf{g}_{i,l}$; and 2) the values of $||\mathbf{g}_l||_2$ and $||\mathbf{g}_{i,l}||_2$. Therefore, we obtained that neighborhood samples (the first aspect) and/or samples near the boundary (the second aspect) will have relatively large value of $||\mathcal{K}^t(\mathbf{x}, \mathbf{x}_i)||_F$. A special sample type with both neighborhood and boundary characteristics is the adversarial example. Appendix A.2 provides theoretical justification. Existing studies have demonstrated the usefulness of adversarial examples in unlearning (Cha et al., 2024), for which we provide a theoretical explanation from data-centric perspective. Additionally, the TOFU dataset (Maini et al., 2024) mainly contains neighborhood data.

Considering that the decision (Softmax) layers of DNNs are essentially linear, for this reason, by summarizing the above analysis, the retain samples should be selected with more priority in the neighborhood set of unlearning samples and the samples near the decision boundary.

## 3 METHODOLOGY

Section 2.2 discusses the optimal label assignment for retain data and which sample should be selected as retain samples according to (5). This section first discusses optimal label assignment for forget samples, then introduces retain sample optimization, and finally gives the entire algorithm.

### 3.1 LABEL ASSIGNMENT OPTIMIZATION

We first introduce the concept of "Unlearning degree", to quantify how much users desire a sample's original output to be unlearned (or forgotten) in a machine learning model. Taking a $C$-class classification problem for example, assume the current model correctly classifies an unlearning sample into class $c$. The unlearning degree can then be operationalized by whether there exist at least $k$ categories with prediction quantities larger than that on $c$. Specifically, if the user only requires removal from top-1 prediction, it suffices to ensure at least one class has a larger predicted probability in the new model than that of $c$. Conversely, if removal from top-2 predictions is desired, there exist at least two classes with larger predicted probabilities in the new model than that of $c$. Crucially, a larger $k$ corresponds to a higher unlearning degree, reflecting a greater forgetting extent. For an unlearning sample $\mathbf{x}$, let $c^* = \arg\max_c [\pi_{\boldsymbol{\theta}^t}(\cdot|\mathbf{x})]_c$. Let $\epsilon$ be a small number satisfying $\epsilon \leq \frac{1}{C-1}$. Consequently, we define the following unlearning degree-aware label assignment objective for $\mathbf{x}$:

$$\mathbf{y}^* = arg\min_{\mathbf{y}} ||\pi_{\boldsymbol{\theta}^t}(\cdot \mid \mathbf{x}) - \mathbf{y}||_2, \text{ s.t. } \mathbf{y} \in \Delta^C, |\{c \neq c^* \mid \mathbf{y}_c \geq \mathbf{y}_{c^*} + \epsilon\}| = k < C. \quad (9)$$

This non-convex problem can be simplified by fixing the category set $S = \{c : \mathbf{y}_c \geq \mathbf{y}_{c^*}\}$, under which a feasible solution is derived in Appendix A.3. Denoting $f = \pi_{\boldsymbol{\theta}^t}(\cdot \mid \mathbf{x})$, we solve the problem in two special cases: For $\boldsymbol{k = 1}$, let $\hat{c} = arg\max_{c \neq c^*}[\pi_{\boldsymbol{\theta}^t}(\cdot \mid \mathbf{x})]_c$. Then:

$$\mathbf{y}_{c^*} = \frac{f_{c^*} + f_{\hat{c}} - \epsilon}{2}, \quad \mathbf{y}_{\hat{c}} = \frac{f_{c^*} + f_{\hat{c}} + \epsilon}{2}, \quad \mathbf{y}_c = f_c \quad \forall c \neq c^*, \hat{c}. \quad (10)$$

For $\boldsymbol{k = C - 1}$, sort the non-$c^*$ probabilities in ascending order: $f_{(1)} \leq \cdots \leq f_{(C-1)}$. Then:

$$\mathbf{y}_{c^*}^* = a_m, \quad \mathbf{y}_c^* = \begin{cases} a_m + \epsilon & \text{if } c \text{ is among the } m \text{ smallest } f_c \ (c \neq c^*) \\ f_c & \text{otherwise} \end{cases}, \quad (11)$$

where $a_m = \frac{1 - m\epsilon - \sum_{i=m+1}^{C-1} f_{(i)}}{m+1}$ and $m \in \{0, 1, \ldots, C-1\}$ is the unique integer satisfying $f_{(m)} < a_m + \epsilon \leq f_{(m+1)}$, $a_m \geq 0$, and $\epsilon \leq \frac{1}{C-1}$. Existence and uniqueness of $m$ are established in the Appendix A.3, along with illustrative examples. This analysis also highlights a limitation of the

classical gradient ascent method (Yao et al., 2024): its implied target (i.e., $\mathbf{y}_i$) is clearly not the optimal solution to (9). Compared with random assignment, the labels from the improved schemes proposed in Zhou et al. (2025); Yang (2025) are closer to our optimization objective. However, these labels are still not optimal solutions either.

**Extension to LLM unlearning**. The presented label assignment framework generalizes to LLMs, where unlearning samples correspond to token sequences and let $\mathbf{r}^*$ represent the dominant output token distribution. The basic optimization framework for current LLM unlearning can be formally defined as finding a new response $\mathbf{r}'$ to solve the following problem:

$$\min_{\mathbf{r}'} \alpha \cdot \text{dist}(\mathbf{r}', \mathbf{r}^*) + \beta \cdot \text{rank}_{\mathcal{S}}(\mathbf{r}') \quad \text{s.t. fluency}(\mathbf{r}') \geq \tau, \quad \text{relevance}(\mathbf{r}', \mathbf{x}) \geq \delta, \quad (12)$$

where $\mathcal{S}$ is the set of sensitive information; $\text{rank}_{\mathcal{S}}(\mathbf{r}')$ represents the inverse rank of the sensitive terms in $\mathbf{r}'$; $\alpha$, $\beta$, $\tau$ and $\delta$ are hyper-parameters. $\beta$ is actually the unlearning degree. Although this framework is conceptually reasonable, there remains significant room for optimization in its core components at the implementation level. This objective is nearly intractable for directly solving under the LLMs scenario. We adopt an approximate optimization scheme consisting of two stages.

- Stage 1: *Generate multiple revised versions of the original response $\mathbf{r}^*$ with an LLM, maintaining semantic coherence while systematically excluding any content pertaining to $S$;*
- Stage 2: *Evaluate each candidate response and select the optimal one as $\mathbf{r}'$. In this study, the scoring criteria is: $\alpha \cdot dist(\mathbf{r}', \mathbf{r}^*) + \gamma_1 \cdot \max(0, \tau - fluency(\mathbf{r}')) + \gamma_2 \cdot \max(0, \delta - relevance(\mathbf{r}', \mathbf{x}))$, where the three functions are implemented by the employed LLM directly.*

Here is an example: Consider a scenario where a model is prompted with "*Explain COVID-19 treatment options*" and initially responds with "*Primary treatments include **remdesivir**, oxygen therapy, and $\cdots$*", disclosing sensitive drug information $\mathcal{S} = \{remdesivir\}$. The sanitized response becomes: "*COVID-19 management primarily focuses on symptomatic care, including respiratory support, fluid balance maintenance, and complication prevention. Certain antiviral agents may be considered in specific cases under medical supervision*". This revision eliminates explicit drug names while preserving clinical utility, whereas current methods usually utilize "IDK" as response. The algorithmic steps are presented in A.4.

### 3.2 RETAIN SET OPTIMIZATION

As earlier sections show, an ideal $\mathcal{RS}$ should include mainly samples near unlearning targets, boundary samples, or both. We provide specific strategies for both standard classification and LLMs. Importantly, our method also uses generation to work even without a retain set.

To obtain neighboring samples, we directly select retain samples based on their similarity to the unlearning targets—using mixup or perturbation when unlearning samples are very scarce. For boundary samples in classification tasks, we first locate the nearest categories for each unlearning sample, then retrieve the closest boundary-aligned samples from each nearest category's retain set, again applying mixup or perturbation if the samples are too far from the boundary. For LLMs, directly identifying boundaries is difficult due to their broad capabilities; instead, for both classification and LLMs, we generate adversarial examples from each unlearning sample. As noted earlier, adversarial examples naturally combine neighborhood and boundary traits.

The optimization refers to neighborhood, boundary, and adversarial samples. Although existing studies have used neighborhood or adversarial samples, their approaches remain heuristic (despite some provided partial theoretical support)—unlike our method is theoretically grounded. Boundary data is not referenced in prior work, but has proven useful

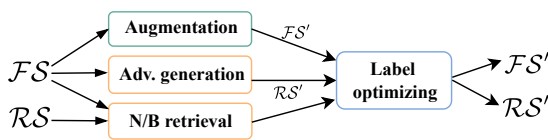

Figure 2: The pipeline of DataOpt.

in our experiments. No prior work has systematically incorporated all three types of retain data.

### 3.3 CONCRETE ALGORITHMS

Building upon the methodologies from the preceding two subsections, we present a unified training data optimization approach (DataOpt) for unlearning as shown in Fig. 2. The algorithmic steps,

---

**Algorithm 1** DataOpt (for classifier or LLM unlearning)

---

**Input**: $\mathcal{FS}$, $\mathcal{RS}$, $\pi_{\boldsymbol{\theta}}$ (a classifier or an LLM), $k$, $k_1$, $k_2$, $\epsilon$, $\mathcal{FS}' = \mathcal{FS}$, $\mathcal{RS}' = null$.
**Output**: $\mathcal{FS}'$, $\mathcal{RS}'$.

1: **for** $i = 1$ **to** $|\mathcal{FS}|$ **do**
2:     Augment $\mathbf{x}_i$ and add generated samples to $\mathcal{FS}'$;
3:     Generate an adversary sample for $\mathbf{x}_i$ and add it to $\mathcal{RS}'$;
4:     Search $k_1$ similar samples for $\mathbf{x}_i$ from $\mathcal{RS}$ and insert them into $\mathcal{RS}'$;
5:     /* *The following two steps are only performed for classification.*
6:     Identify the set $S_c$ of $k_2$ most confusable classes for $\mathbf{x}_i$'s category;
7:     For each class in $S_c$, retrieve from $\mathcal{RS}$ the boundary-aligned sample closest to the boundary between
    that class and $\mathbf{x}_i$'s class, adding the closest one to $\mathbf{x}_i$ of these samples to $\mathcal{RS}'$;
8: **end for**
9: For each sample in $\mathcal{FS}'$, the label is assigned with Eq. (9)/the heuristic scheme for classifier/LLM;
10: For each sample in $\mathcal{RS}'$, the label is assigned with $\pi_{\boldsymbol{\theta}}(\cdot|\mathbf{x})$;
11: **return** $\mathcal{FS}'$, $\mathcal{RS}'$.

---

which systematically integrates both classifier and LLM unlearning tasks, are shown in Algorithm 1.

## 4 EXPERIMENTS

In this section, we empirically validate our proposed data-centric unlearning framework, DataOpt. Our goal is to demonstrate that optimizing the unlearning data itself is a critical and effective strategy. Due to space constraints, we present our main findings here and provide extended results and implementation details in the appendix. The code is provided in the supplementary attachment.

### 4.1 EXPERIMENTAL SETUP

**Tasks and Datasets.** Our evaluation spans two primary unlearning domains: image classifiers and LLMs. **For image classifiers**, we focus on the practical and widely studied scenario of **random subset unlearning**. Following the experimental setup of recent benchmark works (Huang et al., 2024), we conduct experiments on two standard benchmarks: **CIFAR-10** (Krizhevsky et al., 2009) and **Tiny-ImageNet** (Le & Yang, 2015). In both cases, the task is to unlearn a randomly selected 10% subset of the training data. This setup simulates real-world data removal requests under privacy regulations. **For LLMs**, we pivot to the more challenging and practical task of **real-world knowledge unlearning** using the **RWKU benchmark** (Jin et al., 2024). Unlike benchmarks based on fictitious knowledge, RWKU simulates the removal of deeply embedded information about real-world entities. Our task is to perform **batch-target unlearning**, simultaneously removing knowledge of 10, 30, and 50 famous people to test the scalability and robustness of the unlearning methods. We follow the experimental settings of Jin et al. (Jin et al., 2024), including the use of the Llama-3-8B model, the same evaluation probes, and the standardized data generation templates.

**Evaluation Metrics.** We adopt comprehensive metrics to evaluate unlearning performance from different perspectives. **For image classification unlearning**, we measure four key quantitative metrics. To assess model utility preservation and generalizability, we measure (1) **Retain Accuracy (Acc$_{\mathbf{rt}}$ ↑)**, accuracy on the held-out test set of *retained* classes, and (2) **Test Accuracy (Acc$_{\mathbf{test}}$ ↑)**, accuracy on the full test set. To evaluate unlearning efficacy and privacy, we measure (3) **Forget Accuracy (Acc$_{\mathbf{ft}}$ ↓)**, accuracy on the held-out test set of *forgotten* classes, and (4) **MIA (↓)**, the success rate of a Membership Inference Attack on the forget set, quantifying privacy leakage risk (Shokri et al., 2017). An ideal unlearning algorithm should maintain high $Acc_{rt}$ and $Acc_{test}$ while reducing $Acc_{ft}$ and MIA to levels of a model retrained from scratch. **For LLM unlearning**, we adopt the comprehensive RWKU benchmark evaluation framework, assessing performance across three critical dimensions: (1) **Forget Quality (FQ ↑)**, a composite score measuring knowledge removal efficacy. Derived from performance on the *forget set*, including fill-in-the-blank, question-answering, and adversarial attack probes to test unlearning depth and robustness. (2) **Model Utility (MU ↑)**, an aggregate score evaluating preservation of the model's general capabilities. Measured on the *Utility Set*, encompassing standard benchmarks like MMLU (Hendrycks et al., 2021), BBH (Suzgun et al., 2023), and TruthfulQA (Lin et al., 2022). (3) **Relative Utility Drop (RUD ↓)**, a core

metric for quantifying collateral damage. It calculates performance degradation on the *Neighbor Set*—knowledge closely related but not targeted for unlearning—providing a precise measure of the unlearning method's locality. An ideal algorithm should maximize FQ and MU while minimizing RUD's absolute value.

**Baselines.** We compare our method against a wide range of baselines. **Performance Boundaries** are established by the **Original Model** (no unlearning) and the **Retrain Model**. **For image classification unlearning**, we select representative SOTA methods including **Negative Gradient (NEGGRAD)** (Golatkar et al., 2020), **Bad Teacher (BT)** (Chundawat et al., 2023), **SCRUB** (Kurmanji et al., 2023), and **SalUn** (Fan et al., 2023). The **DELETE** method (Zhou et al., 2025), a recent data-centric optimization technique, is also included for comparison. We will show how DataOpt can enhance these methods. **For LLM unlearning**, we select strong and representative baselines including **Gradient Ascent (GA)** (Golatkar et al., 2020), **Negative Preference Optimization (NPO)** (Zhang et al., 2024), and **Rejection Tuning (RT)** (Maini et al., 2024). These methods cover a diverse range of current unlearning strategies. While we do not directly compare against optimization methods that use only a single component (e.g., only neighborhood or adversarial samples), our ablation analysis systematically demonstrates the individual contribution and effectiveness of each of these components within our unified DataOpt framework. The specific hyperparameters and other implementation details for our experiments are presented in Appendix B.

### 4.2 Main Results

**DataOpt Enhances Image Classification Unlearning.** To validate our primary claim, we apply DataOpt as a data-optimization layer to four SOTA unlearning algorithms across two benchmarks, with results presented in Table 1. The data reveals two key findings. First, our framework provides a **superior and more versatile data-optimization strategy** compared to existing methods like DELETE (Zhou et al., 2025). Unlike DELETE which focuses solely on the retain set, our holistic approach optimizes both the retain and forget sets. This leads to a clear advantage across the board: our strong unlearning mode—where $k = C - 1$ (i.e., $k = 9$)—achieves the highest forgetting efficacy (lowest $Acc_{ft}$ and MIA) while maintaining a highly competitive model utility. Simultaneously, our gentle mode ($k = 1$) establishes a new state-of-the-art in utility preservation—achieving near-perfect Retain Accuracy ($Acc_{rt} = 100.00\%$ for SalUn and BT)—while still delivering forgetting performance on par with or superior to DELETE.

Second, and more importantly, DataOpt introduces **controllable unlearning** via the degree parameter $k$. This critical capability, absent in all baselines, allows practitioners to better navigate the utility-privacy trade-off. By selecting $k$, users can tailor the unlearning process to their specific needs, transforming it from a rigid procedure into a versatile tool for tasks ranging from minor model corrections to complete, privacy-critical data removal.

**DataOpt Excels in Real-World LLM Unlearning.** To test our paradigm in a more complex and practical setting, we shift our evaluation to the **RWKU benchmark**. This benchmark simulates a challenging real-world scenario where unlearning must be performed without access to the original training data. Instead, the algorithm is only provided with the target concepts to be forgotten (e.g., the names of 10, 30, or 50 public figures). In this challenging setting where no initial data is provided, the creation and optimization of effective unlearning data becomes a critical first step. Our DataOpt framework is specifically designed for this scenario: we first generate a *proxy forget corpus* by prompting the original model, and then construct a high-value *retain set* using a public corpus and adversarial generation. These datasets are then optimized using our proposed methods. A detailed description of this entire data construction process is provided in Appendix C. We apply DataOpt to three diverse baselines (GA, NPO, and RT) to evaluate its scalability. The results are presented in Table 2.

The data reveals that as the number of forgotten entities increases, vanilla methods suffer from a catastrophic collapse in performance, evidenced by the sharp decline in Model Utility (MU) and the severe collateral damage to related knowledge (rapidly worsening RUD). This is because their indiscriminate unlearning process creates compounding negative interference.

In stark contrast, DataOpt demonstrates remarkable robustness and precision. The primary reason is its **strategic data selection**, which acts as a crucial stabilizing mechanism. By constructing a

Table 1: Enhancement of SOTA unlearning methods via DataOpt on CIFAR-10 and Tiny-ImageNet (10% random subset unlearning). Bold values indicate the best performance.

| Method | CIFAR-10 (10% Random Subset) | | | | Tiny-ImageNet (10% Random Subset) | | | |
|---|---|---|---|---|---|---|---|---|
| | $Acc_{rt} \uparrow$ | $Acc_{ft} \downarrow$ | $Acc_{test} \uparrow$ | **MIA** $\downarrow$ | $Acc_{rt} \uparrow$ | $Acc_{ft} \downarrow$ | $Acc_{test} \uparrow$ | **MIA** $\downarrow$ |
| Original Model | 100.00 | 100.00 | 95.62 | 99.55 | 99.55 | 99.55 | 85.80 | 98.15 |
| Retrain Model | 100.00 | 95.62 | 95.34 | 74.84 | 99.55 | 85.29 | 85.49 | 69.30 |
| NEGGRAD | 94.51 | 99.17 | 88.56 | 76.50 | 86.23 | 91.54 | 72.32 | 78.55 |
| NEGGRAD + DELETE | 94.83 | 98.95 | 88.85 | 75.82 | 86.71 | 91.26 | 72.85 | 77.93 |
| **NEGGRAD + DataOpt (k=1)** | **95.25** | 97.81 | **91.43** | 75.54 | **87.32** | 91.17 | **74.15** | 77.81 |
| **NEGGRAD + DataOpt (k=9)** | 95.02 | **96.82** | 90.65 | **74.95** | 86.54 | **88.53** | 72.50 | **74.82** |
| SalUn | 99.99 | 100.00 | 94.89 | 77.54 | 98.60 | 95.78 | 83.63 | 81.18 |
| SalUn + DELETE | 99.99 | 99.85 | 94.95 | 77.13 | 98.71 | 95.52 | 83.85 | 79.54 |
| **SalUn + DataOpt (k=1)** | **100.00** | 99.45 | **95.12** | 76.21 | **98.85** | 94.53 | **84.32** | 79.15 |
| **SalUn + DataOpt (k=9)** | 99.99 | **97.90** | 94.75 | **75.83** | 98.65 | **93.14** | 83.60 | **77.11** |
| Bad Teacher (BT) | 99.99 | 98.88 | 94.63 | 81.77 | 97.82 | 93.22 | 83.04 | 77.53 |
| BT + DELETE | 99.99 | 98.53 | 94.71 | 80.24 | 97.92 | 92.81 | 83.25 | 76.20 |
| **BT + DataOpt (k=1)** | **100.00** | 98.15 | **94.91** | 79.53 | **98.14** | 91.82 | **83.95** | 76.16 |
| **BT + DataOpt (k=9)** | 99.98 | **97.12** | 94.55 | **77.91** | 97.80 | **90.54** | 83.10 | **73.52** |
| SCRUB | 99.88 | 99.44 | 94.13 | 87.43 | 98.10 | 97.23 | 82.74 | 81.32 |
| SCRUB + DELETE | 99.90 | 99.41 | 94.23 | 85.52 | 98.22 | 97.14 | 83.03 | 79.80 |
| **SCRUB + DataOpt (k=1)** | **99.95** | 99.05 | **94.41** | 84.14 | **98.35** | 96.61 | **83.62** | 78.53 |
| **SCRUB + DataOpt (k=9)** | 99.94 | **97.53** | 94.05 | **80.81** | 98.34 | **94.82** | 82.80 | **74.91** |

Table 2: Enhancement of SOTA LLM unlearning methods via DataOpt on the RWKU benchmark with Llama-3-8B. Best results for each baseline group are in **bold**.

| Method | Forget 10 Entities | | | Forget 30 Entities | | | Forget 50 Entities | | |
|---|---|---|---|---|---|---|---|---|---|
| | FQ $\uparrow$ | MU $\uparrow$ | RUD $\downarrow^{*}$ | FQ $\uparrow$ | MU $\uparrow$ | RUD $\downarrow^{*}$ | FQ $\uparrow$ | MU $\uparrow$ | RUD $\downarrow^{*}$ |
| Original Model | 0.152 | 0.688 | 0.0% | 0.152 | 0.688 | 0.0% | 0.152 | 0.688 | 0.0% |
| GA | 0.751 | 0.624 | -15.3% | 0.812 | 0.591 | -22.1% | 0.855 | 0.553 | -28.4% |
| **GA + DataOpt** | **0.825** | **0.671** | **-2.8%** | **0.863** | **0.662** | **-4.5%** | **0.881** | **0.640** | **-7.2%** |
| NPO | 0.783 | 0.635 | -13.5% | 0.840 | 0.602 | -19.8% | 0.872 | 0.570 | -25.6% |
| **NPO + DataOpt** | **0.831** | **0.675** | **-2.5%** | **0.879** | **0.668** | **-3.9%** | **0.903** | **0.651** | **-6.1%** |
| RT | 0.882 | 0.581 | -18.9% | 0.915 | 0.530 | -26.2% | 0.934 | 0.495 | -33.7% |
| **RT + DataOpt** | **0.905** | **0.668** | **-3.1%** | **0.926** | **0.655** | **-5.3%** | **0.941** | **0.632** | **-8.8%** |

$^{*}$ For RUD, the arrow ($\downarrow$) indicates that less collateral damage (a value closer to 0) is better.

compact, high-value retain set of neighbor and adversarial samples, DataOpt builds a "knowledge anchor" that protects the model's crucial knowledge structure. For instance, when unlearning 50 entities, `GA + DataOpt` reduces the catastrophic -28.4% RUD of the baseline to a minimal -7.2%—a nearly **4-fold reduction in collateral damage**. This trend of exceptional locality preservation and utility maintenance holds true across all baselines and scales. The results provide compelling evidence that in complex, interconnected knowledge systems like LLMs, a data-centric approach is not just beneficial but **fundamental** to achieving safe, scalable, and precise unlearning.

## 4.3 FRAMEWORK ANALYSIS

This section presents our main experimental findings on the key components of our proposed DataOpt, while a comprehensive suite of supplementary experiments is provided in the appendices for further details. These include a computational overhead analysis in Appendix D.1, and a validation of our LLM response generation strategy in Appendix D.2.

**Ablation Study.** We evaluate three core strategies: strategic retain set selection, strategic retain set selection with only boundary samples[2], and optimized forget labels. Table 3 presents the results when these individual and combinations are used. Each component independently improves performance—retain data optimization enhances utility metrics, while forget label optimization improves forgetting efficacy. Notably, our method with forget-label optimization outperforms the DELETE baseline, which also optimizes forget labels. With all components integrated, the full DataOpt framework achieves the best balance, confirming our holistic data-centric design's effectiveness.

---

[2]The effectiveness of neighborhood and adversarial samples has been verified in previous studies(Cha et al., 2024; Li et al., 2025). Therefore, only boundary samples are considered in our ablation study.

Table 3: Ablation study and component-wise analysis of DataOpt on CIFAR-10 (10% forget), built upon the NEGGRAD baseline.

| Method (on top of NEGGRAD) | $Acc_{rt} \uparrow$ | $Acc_{ft} \downarrow$ | $Acc_{test} \uparrow$ | MIA $\downarrow$ |
|---|---|---|---|---|
| Baseline (Random Retain + GT Labels) | 94.51 | 99.17 | 88.56 | 76.50 |
| + DELETE (Retain Set) | 94.83 | 98.95 | 88.85 | 75.82 |
| + DataOpt Retain Set only | **94.90** | **97.20** | **89.05** | **75.80** |
| + DataOpt Boundary sample only | **94.90** | **97.20** | **89.05** | **75.80** |
| + DataOpt Forget Labels only | **94.75** | **97.51** | **89.22** | **75.13** |
| Full DataOpt (k=9)) | **95.02** | **96.82** | **90.65** | **74.95** |

Table 4: Impact of unlearning degree (k) on CIFAR-10, applied to the NEGGRAD baseline.

| k | $Acc_{rt} \uparrow$ | $Acc_{ft} \downarrow$ | $Acc_{test} \uparrow$ | MIA $\downarrow$ |
|---|---|---|---|---|
| 1 | 95.25±0.2 | 97.81±0.4 | 91.43±0.3 | 75.54±1.2 |
| 2 | 95.17±0.2 | 97.76±0.4 | 91.34±0.3 | 75.31±1.2 |
| 3 | 95.05±0.2 | 97.72±0.6 | 91.15±0.3 | 75.23±1.5 |
| 5 | 94.98±0.3 | 97.58±0.8 | 90.93±0.4 | 75.16±1.8 |
| 7 | 94.82±0.3 | 97.13±0.9 | 90.85±0.4 | 75.03±2.1 |
| 9 (C − 1) | 95.02±0.3 | 96.82±1.1 | 90.65±0.4 | 74.95±2.4 |

**Controllability of Unlearning.** A unique feature of DataOpt is its ability to control the unlearning intensity via the degree parameter $k$. To demonstrate this, we apply DataOpt with varying degrees of $k$ to the NEGGRAD baseline on CIFAR-10. As shown in Table 4, as $k$ increases from a gentle unlearning setting ($k = 1$) to a complete one ($k = 9$), both the forget accuracy ($Acc_{ft}$) and the MIA success rate drop significantly and smoothly, demonstrating stronger and more private unlearning. Crucially, both the retain accuracy ($Acc_{rt}$) and the overall test accuracy ($Acc_{test}$) remain highly stable with only minor, graceful degradation. This level of granular control, combined with the principled and consistent hyperparameter settings used for our framework (detailed in Appendix C), underscores its robustness and adaptability to diverse unlearning requirements.

## 5 RELATED WORK

Machine unlearning research spans traditional models and large language models (LLMs), predominantly emphasizing algorithmic and architectural innovations over data-centric strategies such as BT (Chundawat et al., 2023), SCRUB (Kurmanji et al., 2023), SalUn (Fan et al., 2023), NPO (Zhang et al., 2024), and ICU (Pawelczyk et al., 2024). Recently, researchers have begun to identify issues in current training data setups for machine unlearning. Zhou et al. (2025) examined the drawback of using random one-hot labels in classifier unlearning, noting its negative impact on randomly assigned classes. Yuan et al. (2025) pointed out that using IDK as the unlearning target causes model outputs for normal data to also become biased toward such responses. In response, some studies optimize unlearning targets through generation or regularization to mitigate the effect of fixed unlearning labels (Yang, 2025; Zhou et al., 2025; Di et al., 2024). Other researchers have focused on improving the retain set—for instance, Li et al. (2025) selected retain samples based on similarity to forget set samples, reflecting the idea that neighborhood data makes more effective retain samples. Some also adopted adversarial generation to create adversarial examples (Chen et al., 2025). Overall, most of these efforts are heuristic-driven, though some are theoretically supported. A unified theoretical framework from a pure data perspective for optimizing labels and retain data is lacking.

Data-centric machine learning has gained increasing attention in recent years. In both conventional deep learning and LLM research, improving final model performance from a data perspective has become a mainstream paradigm (Wu & Yao, 2025; Wu, 2025). In traditional deep learning, common data enhancement strategies include sample perturbation (e.g., adversarial examples), augmentation (e.g., neighborhood mixup), label perturbation (e.g., label smoothing), selection (e.g., gradient-based (Calian et al., 2025)), and reweighting. These schemes are increasingly being adopted in machine unlearning as well. In the LLM domain, sample selection (e.g., Xia et al. (2024) and Chhabra et al. (2024)) and augmentation (Ding et al., 2024) are two widely used data improvement strategies, which are also being introduced into LLM unlearning. We believe that data-centric unlearning will gradually become a main research paradigm in machine unlearning.

## 6 CONCLUSIONS

This work addresses the critical gap in training data construction for machine unlearning by introducing learning dynamics theory. We derive two key principles: (1) an optimization function for label assignment to guide effective forgetting, and (2) the strategic selection of neighborhood/boundary samples as optimal retain data. Implementing these principles for both classifiers and LLMs, our data-centric strategy, namely DataOpt, consistently enhances existing unlearning algorithms across diverse tasks (both image classifier and LLM unlearning). Ablation and sensitivity analyses confirm the robustness of the entire framework and the effectiveness of each key component. This establishes deliberate data optimization as fundamental to high-performance unlearning.

ETHICS STATEMENT

Our research focuses on advancing unlearning algorithms, with the sole goal of improving model performance on publicly available datasets. All experimental data are from open repositories (details in main text/supplements) and comply with open-source licenses, requiring no additional ethical clearance for academic use. This study adheres to academic ethical guidelines and responsible AI development principles, with no direct harm to individuals or groups.

REPRODUCIBILITY STATEMENT

All experiments in this work are conducted on publicly available datasets, whose sources are detailed in the main text and supplementary materials. All custom algorithm implementations and experimental code have been included in the supplementary appendix. Upon acceptance of this paper, we will make the complete codebase publicly available on GitHub, with detailed documentation on environment setup, hyperparameter configurations, and step-by-step instructions to reproduce the reported results. We ensure that all key experimental findings can be replicated using the provided data and code.

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

## USE OF LARGE LANGUAGE MODELS

Consistent with ICLR policy, we report that LLMs were used as writing assistants in the paper preparation. Their primary role was to assist with grammar correction and language polishing, with the aim of enhancing the readability of the text. All core ideas and analyses were developed by the human authors, who bear full responsibility for the final content of the paper.

## A   THEORETICAL ANALYSIS

### A.1   PROOF FOR LEMMA 1 AND MULTI-CLASS EXTENSION

**Proof 1** *Recall that for binary classification, the model output $\pi_{\boldsymbol{\theta}^t}(\cdot \mid \mathbf{x})$ is a probability vector $[p, q]^\top$ where $q = 1 - p$. The term $\mathcal{A}^t(\mathbf{x})$ is defined as:*

$$\mathcal{A}^t(\mathbf{x}) = \nabla_{\mathbf{z}} \log \pi_{\boldsymbol{\theta}^t}(\mathbf{x}) = I - 1\pi_{\boldsymbol{\theta}^t}^\top(\cdot \mid \mathbf{x}),$$

*which for binary classification simplifies to the matrix:*

$$\mathcal{A}^t(\mathbf{x}) = \begin{pmatrix} q & -q \\ -p & p \end{pmatrix}.$$

*Its Frobenius norm is given by:*

$$\|\mathcal{A}^t(\mathbf{x})\|_F = \sqrt{q^2 + (-q)^2 + (-p)^2 + p^2} = \sqrt{2p^2 + 2q^2} = \sqrt{2(p^2 + q^2)}. \tag{13}$$

*Since $q = 1 - p$, we have $p^2 + q^2 = p^2 + (1 - p)^2 = 2p^2 - 2p + 1$, and therefore:*

$$\|\mathcal{A}^t(\mathbf{x})\|_F = \sqrt{2(2p^2 - 2p + 1)} = \sqrt{4p^2 - 4p + 2}. \tag{14}$$

*The $L_2$ norm of the log probability vector is:*

$$\| \log \pi_{\boldsymbol{\theta}^t}(\cdot \mid \mathbf{x})\|_2 = \sqrt{(\log p)^2 + (\log q)^2}. \tag{15}$$

*Thus, the ratio $R(p)$ becomes:*

$$R(p) = \frac{\|\mathcal{A}^t(\mathbf{x})\|_F}{\| \log \pi_{\boldsymbol{\theta}^t}(\cdot \mid \mathbf{x})\|_2} = \frac{\sqrt{4p^2 - 4p + 2}}{\sqrt{(\log p)^2 + (\log(1 - p))^2}}. \tag{16}$$

*At the decision boundary $p = 0.5$:*

$$\|\mathcal{A}^t(\mathbf{x})\|_F = \sqrt{4(0.5)^2 - 4(0.5) + 2} = \sqrt{1 - 2 + 2} = 1, \tag{17}$$

$$\| \log \pi_{\boldsymbol{\theta}^t}(\cdot \mid \mathbf{x})\|_2 = \sqrt{2(\log 0.5)^2} = \sqrt{2}|\log 0.5|, \tag{18}$$

$$R(0.5) = \frac{1}{\sqrt{2}|\log 0.5|} = \frac{1}{\sqrt{2}\log 2}. \tag{19}$$

*As $p \to 0$ or $p \to 1$:*

$$\|\mathcal{A}^t(\mathbf{x})\|_F \to \sqrt{2}, \tag{20}$$

$$\|\log \pi_{\boldsymbol{\theta}^t}(\cdot \mid \mathbf{x})\|_2 \to \infty, \tag{21}$$

$$R(p) \to 0. \tag{22}$$

*Now consider the squared ratio:*

$$S(p) = [R(p)]^2 = \frac{4p^2 - 4p + 2}{(\log p)^2 + (\log(1-p))^2}. \tag{23}$$

*Let $N(p) = 4p^2 - 4p + 2$ and $D(p) = (\log p)^2 + (\log(1-p))^2$, so $S(p) = N(p)/D(p)$. The derivative is:*

$$S'(p) = \frac{N'(p)D(p) - N(p)D'(p)}{[D(p)]^2}, \tag{24}$$

*where:*

$$N'(p) = 8p - 4, \quad D'(p) = \frac{2\log p}{p} - \frac{2\log(1-p)}{1-p}. \tag{25}$$

*For $p \in (0, 0.5)$, the numerator $T(p) = N'(p)D(p) - N(p)D'(p) > 0$, hence $S'(p) > 0$, indicating that $S(p)$ is monotonically increasing on $(0, 0.5)$. By symmetry, $S'(p) < 0$ for $p \in (0.5, 1)$, so $S(p)$ is monotonically decreasing on $(0.5, 1)$. Therefore, $S(p)$ attains its maximum at $p = 0.5$, and consequently $R(p)$ also attains its maximum at $p = 0.5$.*

*Thus, the ratio $\frac{\|\mathcal{A}^t(\mathbf{x})\|_F}{\|\log \pi_{\boldsymbol{\theta}^t}(\cdot|\mathbf{x})\|_2}$ increases monotonically as $\mathbf{x}$ approaches the decision boundary and is maximized when $\mathbf{x}$ lies on the boundary.*

For multi-class classification problems, the decision boundaries can be quite complex. They may form surfaces between two classes, or involve three or even more classes. Generally speaking, when the top-$h$ largest components of a sample's softmax output are equal or nearly equal, that sample lies near the decision boundary between those $h$ classes. We therefore have the following lemma:

**Lemma 2** *For a given sample $\mathbf{x}$ in a $C$-class linear classification problem with softmax output, as the top $h$ probability components approach uniformity while the other components remain fixed, the ratio $\frac{\|\mathcal{A}^t(\mathbf{x})\|_F}{\|\log \pi_{\boldsymbol{\theta}^t}(\cdot|\mathbf{x})\|_2}$ attains its maximum when the top $h$ components are equal.*

**Proof 2** *Let $\mathbf{p} = \pi_{\boldsymbol{\theta}^t}(\cdot \mid \mathbf{x})$ be the probability vector with components $p_i$ for $i = 1, \ldots, C$. Let $S$ be the set of indices of the top $h$ components, and let $s = \sum_{i \in S} p_i$ be constant since the other components are fixed. We consider the variation of $p_i$ for $i \in S$ under the constraint $\sum_{i \in S} p_i = s$.*

*From the given formula,*

$$\mathcal{A}^t(\mathbf{x}) = I - \mathbf{1}\mathbf{p}^\top, \tag{26}$$

*where $I$ is the $C \times C$ identity matrix and $\mathbf{1}$ is the all-ones vector. The Frobenius norm squared is:*

$$\|\mathcal{A}^t(\mathbf{x})\|_F^2 = \|I - \mathbf{1}\mathbf{p}^\top\|_F^2 = \sum_{i=1}^{C}\sum_{j=1}^{C}(\delta_{ij} - p_j)^2 = C - 2 + C\|\mathbf{p}\|_2^2, \tag{27}$$

*as derived from element-wise calculation.*

*The denominator is:*

$$\|\log \mathbf{p}\|_2^2 = \sum_{i=1}^{C}(\log p_i)^2. \tag{28}$$

*For $i \notin S$, $p_i$ are fixed, so let $K_1 = \sum_{i \notin S}(\log p_i)^2$ and $K_2 = \sum_{i \notin S} p_i^2$ be constants. Then:*

$$\|\mathbf{p}\|_2^2 = \sum_{i \in S} p_i^2 + K_2, \quad \|\log \mathbf{p}\|_2^2 = \sum_{i \in S}(\log p_i)^2 + K_1. \tag{29}$$

*Thus, the ratio squared becomes:*

$$R^2 = \frac{\|\mathcal{A}^t(\mathbf{x})\|_F^2}{\|\log \mathbf{p}\|_2^2} = \frac{C - 2 + C(\sum_{i \in S} p_i^2 + K_2)}{\sum_{i \in S}(\log p_i)^2 + K_1} = \frac{D + CA}{B + K_1}, \tag{30}$$

*where $D = C - 2 + CK_2$, $A = \sum_{i \in S} p_i^2$, and $B = \sum_{i \in S} (\log p_i)^2$.*

*Under the constraint $\sum_{i \in S} p_i = s$, $A$ is minimized when all $p_i$ for $i \in S$ are equal (by the Cauchy-Schwarz inequality). Similarly, $B$ is minimized when all $p_i$ are equal because the function $f(p) = (\log p)^2$ is strictly convex for $p \in (0, 1]$ (since $f''(p) = 2(1 - \log p)/p^2 > 0$ for $p \in (0, 1]$), and by Jensen's inequality, $\sum_{i \in S} f(p_i) \geq h f(s/h)$ with equality only when all $p_i$ are equal.*

*Although both $A$ and $B$ are minimized at uniformity, the function $f(p) = (\log p)^2$ has greater curvature than $g(p) = p^2$ for $p \in (0, 1]$. This means that when the $p_i$ deviate from equality, $B$ increases more rapidly than $A$. Consequently, the denominator $B + K_1$ increases faster than the numerator $D + CA$, causing $R^2$ to decrease. Therefore, $R^2$ attains its maximum when the top $h$ components are equal, completing the proof.*

Lemma 2 also indicates that samples located on the decision boundary have a higher $\frac{\|\mathcal{A}^t(\mathbf{x})\|_F}{\|\log \pi_{\boldsymbol{\theta}^t}(\cdot|\mathbf{x})\|_2}$ value compared to non-boundary samples in their vicinity.

## A.2 THE LOCATION OF ADVERSARIAL SAMPLES

Let us consider a classification function $f : \mathbb{R}^n \to \mathcal{Y}$, where $\mathcal{Y} = \{1, 2, \ldots, K\}$ is a set of class labels. The decision rule is typically $f(\mathbf{x}) = \arg\max_i (S(\mathbf{x}))_i$, where $S : \mathbb{R}^n \to \mathbb{R}^K$ is a scoring or probability function (e.g., softmax output). Given a legitimate sample $\mathbf{x} \in \mathbb{R}^n$ with true label $y$, an adversarial sample $\mathbf{x}^*$ is defined as:

$$\mathbf{x}^* = \mathbf{x} + \boldsymbol{\delta}, \tag{31}$$

$$\text{s.t.} \quad f(\mathbf{x}^*) \neq y, \|\boldsymbol{\delta}\|_p \leq \epsilon, \tag{32}$$

where $\boldsymbol{\delta}$ is an additive perturbation, $\|\cdot\|_p$ is an $\ell_p$-norm, and $\epsilon > 0$ is a small, predefined constant ensuring imperceptibility. Note that the decision boundary between class $i$ and $j$ is the set $\mathcal{B}_{i,j} = \{\mathbf{z} \in \mathbb{R}^n : (S(\mathbf{z}))_i = (S(\mathbf{z}))_j\}$. We have

**Proposition 1** *An adversarial sample $\mathbf{x}^*$ as defined above is: (1) Contained within a local $\epsilon$-neighborhood of $\mathbf{x}$. (2) Located near a decision boundary of the classifier $f$.*

**Proof 3** *We prove both statements sequentially. The constraint $\|\boldsymbol{\delta}\|_p = \|\mathbf{x}^* - \mathbf{x}\|_p \leq \epsilon$ is a direct definition of a closed ball (neighborhood) in the metric space $(\mathbb{R}^n, \|\cdot\|_p)$. Therefore, $\mathbf{x}^*$ is constrained to lie within the set:*

$$\mathcal{N}_p(\mathbf{x}; \epsilon) = \{\mathbf{z} \in \mathbb{R}^n : \|\mathbf{z} - \mathbf{x}\|_p \leq \epsilon\}.$$

*This is, by definition, a local $\epsilon$-neighborhood of the original sample $\mathbf{x}$. Let the original classification be $f(\mathbf{x}) = y$. The adversarial constraint requires $f(\mathbf{x}^*) \neq y$. Let the new classification be $f(\mathbf{x}^*) = y'$ where $y' \neq y$. Consider the continuous scoring function $S$. The prediction change implies that a score component $(S(\mathbf{z}))_{y'}$ has become larger than $(S(\mathbf{z}))_y$ at $\mathbf{z} = \mathbf{x}^*$, whereas the opposite was true (or at least they were ordered differently) at $\mathbf{z} = \mathbf{x}$.*

*Define a continuous path $\phi(\lambda) = \mathbf{x} + \lambda\boldsymbol{\delta}$ for $\lambda \in [0, 1]$ connecting $\mathbf{x}$ to $\mathbf{x}^*$. Since $S$ is continuous, the function $g(\lambda) = (S(\phi(\lambda)))_{y'} - (S(\phi(\lambda)))_y$ is also continuous. We know:*

$$g(0) = (S(\mathbf{x}))_{y'} - (S(\mathbf{x}))_y \leq 0 \quad \text{(since } y \text{ was the argmax at } \mathbf{x}),$$

$$g(1) = (S(\mathbf{x}^*))_{y'} - (S(\mathbf{x}^*))_y > 0 \quad \text{(since } y' \text{ is the argmax at } \mathbf{x}^*).$$

*By the Intermediate Value Theorem, there exists some $\lambda^* \in (0, 1)$ such that $g(\lambda^*) = 0$. The point $\mathbf{z}^* = \phi(\lambda^*)$ lies on the decision boundary $\mathcal{B}_{y,y'}$. Since $\|\mathbf{z}^* - \mathbf{x}\|_p = \|\lambda^*\boldsymbol{\delta}\|_p = \lambda^*\|\boldsymbol{\delta}\|_p \leq \lambda^*\epsilon \leq \epsilon$, the point $\mathbf{z}^*$ on the boundary is also within the $\epsilon$-neighborhood of $\mathbf{x}$. The adversarial sample $\mathbf{x}^*$ itself is a point on the path $\phi(\lambda)$ that is arbitrarily close to this boundary point $\mathbf{z}^*$ (the path is continuous). Therefore, $\mathbf{x}^*$ must be located near the decision boundary $\mathcal{B}_{y,y'}$.*

The standard definition of an adversarial sample intrinsically links it to the concepts of locality and decision boundaries. The norm constraint $\|\delta\|_p \leq \epsilon$ enforces locality, while the misclassification requirement, coupled with the continuity of the underlying model, forces the sample to exist near a region where the model's confidence between two classes is equal—i.e., the decision boundary.

This formal argument confirms the intuitive understanding of adversarial samples and provides a foundational principle for studying their properties and developing defenses. Fig. 3 illustrates an illustration of adversarial examples from Madry et al. (2019), where it can be clearly observed that these adversarial examples reside within both the neighborhood and the vicinity of the decision boundary.

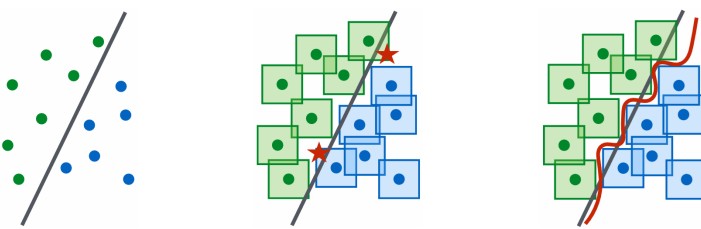

Figure 3: An illustration of adversarial examples (red star) from Madry et al. (2019).

### A.3 THE SOLVING FOR 9

#### A.3.1 THE SOLVING FOR CASE 1: $k = 1$

For $k = 1$, exactly one class $c \neq c^*$ must satisfy $\mathbf{y}_c \geq \mathbf{y}_{c^*} + \epsilon$. Let $\hat{c} = \arg\max_{c \neq c^*} f_c$ where $f = \pi_{\boldsymbol{\theta}^t}(\cdot \mid \mathbf{x})$. The solution can be described by the following proposition:

**Proposition 2** *The optimal solution for $k = 1$ is:*

$$\mathbf{y}_{c^*} = \frac{f_{c^*} + f_{\hat{c}} - \epsilon}{2}, \quad \mathbf{y}_{\hat{c}} = \frac{f_{c^*} + f_{\hat{c}} + \epsilon}{2}, \quad \mathbf{y}_c = f_c \quad \forall c \neq c^*, \hat{c} \tag{33}$$

**Proof 4** *The Lagrangian for this constrained optimization is:*

$$\mathcal{L} = \frac{1}{2}\|\mathbf{f} - \mathbf{y}\|_2^2 + \lambda \left( \sum_{c=1}^{C} y_c - 1 \right) + \mu(y_{\hat{c}} - y_{c^*} - \epsilon) \tag{34}$$

*where we've incorporated the constraint $y_{\hat{c}} \geq y_{c^*} + \epsilon$ with a Lagrange multiplier $\mu \geq 0$. Taking partial derivatives and setting to zero:*

$$\frac{\partial \mathcal{L}}{\partial y_{c^*}} = -(f_{c^*} - y_{c^*}) + \lambda - \mu = 0 \tag{35}$$

$$\frac{\partial \mathcal{L}}{\partial y_{\hat{c}}} = -(f_{\hat{c}} - y_{\hat{c}}) + \lambda + \mu = 0 \tag{36}$$

$$\frac{\partial \mathcal{L}}{\partial y_c} = -(f_c - y_c) + \lambda = 0 \quad \forall c \neq c^*, \hat{c} \tag{37}$$

*From the KKT conditions, we know that either $\mu = 0$ or the constraint is active ($y_{\hat{c}} = y_{c^*} + \epsilon$). Since we require exactly one class to exceed $y_{c^*}$ by at least $\epsilon$, the constraint must be active, giving:*

$$y_{\hat{c}} = y_{c^*} + \epsilon \tag{38}$$

*Solving the system of equations:*

$$y_{c^*} = f_{c^*} - \lambda + \mu \tag{39}$$
$$y_{\hat{c}} = f_{\hat{c}} - \lambda - \mu \tag{40}$$
$$y_c = f_c - \lambda \quad \forall c \neq c^*, \hat{c} \tag{41}$$

*Substituting the active constraint:*

$$f_{\hat{c}} - \lambda - \mu = f_{c^*} - \lambda + \mu + \epsilon \Rightarrow f_{\hat{c}} - f_{c^*} = 2\mu + \epsilon \tag{42}$$

*Also, from the simplex constraint $\sum_c y_c = 1$:*

$$\sum_c f_c - C\lambda - \mu + \mu = 1 \Rightarrow \lambda = \frac{\sum_c f_c - 1}{C} = 0, \tag{43}$$

*since $f$ is already on the simplex. Thus, we have:*

$$\mu = \frac{f_{\hat{c}} - f_{c^*} - \epsilon}{2} \tag{44}$$

$$y_{c^*} = f_{c^*} + \mu = \frac{f_{c^*} + f_{\hat{c}} - \epsilon}{2} \tag{45}$$

$$y_{\hat{c}} = f_{\hat{c}} - \mu = \frac{f_{c^*} + f_{\hat{c}} + \epsilon}{2} \tag{46}$$

*The other components remain unchanged: $y_c = f_c$ for $c \neq c^*, \hat{c}$.*

### A.3.2 THE SOLVING FOR CASE 2: $k = C - 1$

For $k = C - 1$, all classes $c \neq c^*$ must satisfy $\mathbf{y}_c \geq \mathbf{y}_{c^*} + \epsilon$. The solution is described as:

**Proposition 3** *The optimal solution for $k = C - 1$ is:*

$$\mathbf{y}_{c^*}^* = a_m, \quad \mathbf{y}_c^* = \begin{cases} a_m + \epsilon & \text{if } c \text{ is among the } m \text{ smallest } f_c \ (c \neq c^*) \\ f_c & \text{otherwise} \end{cases} \tag{47}$$

*where $a_m = \frac{1 - m\epsilon - \sum_{i=m+1}^{C-1} f_{(i)}}{m+1}$ and $m$ is the unique integer satisfying:*

$$f_{(m)} < a_m + \epsilon \leq f_{(m+1)}, \quad a_m \geq 0 \tag{48}$$

**Proof 5** *Let $S$ be the set of classes $c \neq c^*$ for which we increase $y_c$ to exactly $y_{c^*} + \epsilon$, and let $T$ be the remaining classes that we leave unchanged ($y_c = f_c$). Let $|S| = m$. The optimization problem becomes:*

$$\min_{y_{c^*}, \{y_c\}_{c \in S}} \left[ (f_{c^*} - y_{c^*})^2 + \sum_{c \in S} (f_c - (y_{c^*} + \epsilon))^2 \right] \tag{49}$$

*subject to:*

$$y_{c^*} + \sum_{c \in S} (y_{c^*} + \epsilon) + \sum_{c \in T} f_c = 1 \tag{50}$$

*From the simplex constraint:*

$$(m+1)y_{c^*} + m\epsilon + \sum_{c \in T} f_c = 1 \Rightarrow y_{c^*} = \frac{1 - m\epsilon - \sum_{c \in T} f_c}{m+1} \tag{51}$$

*Let $a_m = y_{c^*}$. The objective function becomes:*

$$(f_{c^*} - a_m)^2 + \sum_{c \in S} (f_c - a_m - \epsilon)^2 \tag{52}$$

*To minimize this expression, we should include in $S$ the classes with the smallest values of $f_c$, as these contribute most to the objective when increased. Thus, we sort the non-$c^*$ probabilities in ascending order: $f_{(1)} \leq f_{(2)} \leq \cdots \leq f_{(C-1)}$. We need to find the largest $m$ such that:*

$$f_{(m)} < a_m + \epsilon \leq f_{(m+1)} \tag{53}$$

*where:*

$$a_m = \frac{1 - m\epsilon - \sum_{i=m+1}^{C-1} f_{(i)}}{m+1} \tag{54}$$

*This condition ensures that: (1) For classes in $S$ (the $m$ smallest), we have $f_c \leq a_m + \epsilon$, so increasing them to $a_m + \epsilon$ decreases the objective. (2) For classes in $T$ (the remaining), we have $f_c \geq a_m + \epsilon$, so leaving them unchanged is optimal.*

*The existence and uniqueness of $m$ can be shown by considering $m$ from $0$ to $C - 1$ and verifying that the conditions hold for exactly one value of $m$. Once $m$ is determined, the optimal solution is:*

$$\mathbf{y}_{c^*}^* = a_m, \quad \mathbf{y}_c^* = \begin{cases} a_m + \epsilon & \text{for } c \in S \\ f_c & \text{for } c \in T \end{cases} \tag{55}$$

To facilitate a more intuitive understanding of our proposed solution, several illustrative examples of problem-solving are presented in Table 5. As observed from the tabulated data, the probability mass associated with the original class of the sample diminishes rapidly as $k$ increases, and this probability also decreases correspondingly when $\epsilon$ increases.

Table 5: Comparison of unlearning label assignment under different $k$ and $\epsilon$ ($C = 5$).

| $\epsilon$ | Initial distribution $f$ | $k = 1$ | $k = 4$ |
|---|---|---|---|
| 0.2 | [0.3, 0.25, 0.2, 0.15, 0.1] | [0.175, 0.375, 0.2, 0.15, 0.1] | [0.0375, 0.25, 0.2375, 0.2375, 0.2375] |
| | [0.5, 0.2, 0.15, 0.1, 0.05] | [0.25, 0.45, 0.15, 0.1, 0.05] | [0.04, 0.24, 0.24, 0.24, 0.24] |
| 0.15 | [0.3, 0.25, 0.2, 0.15, 0.1] | [0.2, 0.35, 0.2, 0.15, 0.1] | [0.075, 0.25, 0.225, 0.225, 0.225] |
| | [0.5, 0.2, 0.15, 0.1, 0.05] | [0.3, 0.45, 0.15, 0.1, 0.05] | [0.0875, 0.2375, 0.2375, 0.2375, 0.2375] |
| 0.1 | [0.3, 0.25, 0.2, 0.15, 0.1] | [0.225, 0.325, 0.2, 0.15, 0.1] | [0.1125, 0.25, 0.2125, 0.2125, 0.2125] |
| | [0.5, 0.2, 0.15, 0.1, 0.05] | [0.3, 0.4, 0.15, 0.1, 0.05] | [0.12, 0.22, 0.22, 0.22, 0.22] |
| 0.05 | [0.3, 0.25, 0.2, 0.15, 0.1] | [0.25, 0.3, 0.2, 0.15, 0.1] | [0.15, 0.25, 0.2, 0.2, 0.2] |
| | [0.5, 0.2, 0.15, 0.1, 0.05] | [0.325, 0.375, 0.15, 0.1, 0.05] | [0.16, 0.21, 0.21, 0.21, 0.21] |

## A.4 ALGORITHM FOR LLM

The construction process for the labels (i.e., responses) of the forget set samples in LLMs unlearning is described in Algorithm 2.

---

**Algorithm 2** New Response Generation for LLM Unlearning

---

**Require:** Original response $\mathbf{r}^*$, sensitive information set $\mathcal{S}$, hyperparameters $\alpha$, $\beta$, $\tau$, $\delta$, $\gamma_1$, $\gamma_2$
**Ensure:** Sanitized response $\mathbf{r}'$
 1: **Stage 1: Response Generation**
 2: **for** $i = 1$ to $N$ **do**
 3:     $\mathbf{r}_i \leftarrow \text{LLM}_{\text{revise}}(\mathbf{r}^*, \mathcal{S})$
 4: **end for**
 5: **Stage 2: Response Selection**
 6: **for** each candidate response $\mathbf{r}_i$ **do**
 7:     $\text{score}_i \leftarrow \alpha \cdot \text{dist}(\mathbf{r}_i, \mathbf{r}^*) + \gamma_1 \cdot \max(0, \tau - \text{fluency}(\mathbf{r}_i)) + \gamma_2 \cdot \max(0, \delta - \text{relevance}(\mathbf{r}_i, \mathbf{x}))$
 8: **end for**
 9: $\mathbf{r}' \leftarrow \arg\min_{\mathbf{r}_i} \text{score}_i$ {Select optimal response minimizing the scoring function}
10: **return** $\mathbf{r}'$

---

# B IMPLEMENTATION AND HYPERPARAMETER DETAILS

This section provides a detailed breakdown of all implementation choices and hyperparameters used in our experiments to ensure full reproducibility.

## B.1 UNLEARNING TRAINING PARAMETERS

- **Hardware:** Image classification experiments were conducted on a single NVIDIA A40 48GB GPU. LLM experiments were conducted on four NVIDIA A40 48GB GPUs.

- **Software:** Our implementation is based on PyTorch 2.1 and Hugging Face Transformers 4.3.0.

- **Optimizer:** We used the AdamW optimizer for all experiments.

- **Training Epochs:** All image classification models were fine-tuned for 5 epochs. All LLM unlearning models were fine-tuned for 3 epochs.

- **Batch Size:** A batch size of 128 was used for image classification tasks. A batch size of 8 was used for LLM unlearning tasks.

- **Learning Rates:** Learning rates for all baseline methods were individually tuned via grid search to find the optimal trade-off between forgetting quality and model utility. The best-performing learning rates used for reporting results are listed in Table 6.

## B.2 DATAOPT FRAMEWORK HYPERPARAMETERS

In this section, we detail the specific hyperparameter choices for our DataOpt framework. These settings were kept consistent across all experiments to provide a fair and robust evaluation of our method's efficacy.

- **For Classifier Tasks (as per Algorithm 1):**

  - **Augmented Samples:** We generated **3** augmented versions for each sample in the forget set using standard transformations (random crop and horizontal flip).
  - **Adversarial Samples:** We generated **1** adversarial sample for each forget sample. The generation was performed using Projected Gradient Descent (PGD) with 10 steps and a step size of 2/255. We chose a single adversarial sample to incorporate crucial boundary information without excessively biasing the training gradient away from the natural data distribution.
  - **Neighborhood Samples ($k_1$):** The number of neighborhood samples was set to **1** ($k_1 = 1$). For each forget sample, we selected its single nearest neighbor from the retain pool based on cosine similarity in the feature space of the pre-trained ResNet-18 model.
  - **Boundary Samples ($k_2$):** The number of boundary samples was set to **1**. We first identified the single most confusable class for the forget sample's true class ($k_2 = 1$) and then selected the boundary-aligned sample from that class that was geometrically closest to the forget sample.
  - **Forget Set Label Generation ($\epsilon$):** The value of $\epsilon$ in Eq. (9), which controls the separation margin in the target label distribution, was set to a small constant of $\frac{1}{2(C-1)}$.

- **For LLM Tasks:**

  - **Candidate Response Generation:** For each proxy response, we generated $N = 5$ candidate responses in Stage 1 of our optimization pipeline. This provided a diverse set of options for the selection stage.
  - **Retain Set Construction:** For each target entity, the retain set was constructed by generating **1** adversarial question designed to probe the model's decision boundary, and retrieving **5** topically related texts about neighboring entities from our Wikipedia corpus.

## C  EXPERIMENTAL SETUP DETAILS

This section provides a comprehensive overview of the experimental setup, including datasets, models, evaluation metrics, and implementation details to ensure full reproducibility of our results.

### C.1  DATASETS AND TASKS

Our evaluation spans two primary domains: image classification and LLMs.

#### C.1.1  IMAGE CLASSIFICATION

For image classification, we conduct experiments on **CIFAR-10** and **Tiny-ImageNet**. We evaluate the *Random Subset Unlearning* scenario, where a random 10% portion of the training data is forgotten. This setup, following recent benchmarks (Huang et al., 2024), simulates data removal requests where both the forget samples and a pool of retain samples are available to the unlearning algorithm.

#### C.1.2  LARGE LANGUAGE MODELS

For LLMs, we shift to the more challenging and realistic task of *real-world knowledge unlearning* using the **RWKU benchmark** (Jin et al., 2024). This benchmark is particularly suited to evaluate our DataOpt framework due to its practical setting where no original training or forget corpus is provided. Instead, the algorithm is only given the target concepts to be unlearned (e.g., the names of 10, 30, or 50 entities). Our task is *batch-target unlearning* at these different scales.

In this context, the primary challenge is to construct the necessary unlearning datasets from scratch. We address this using the following methodology integrated within our DataOpt framework:

**Optimized Forget Set Construction.**  The process begins with generating a **proxy forget corpus**. We prompt the original model with a set of predefined questions about the target entity (e.g., "Who

is Elon Musk?") to elicit factual responses. These responses, containing the knowledge to be erased, are then processed by our two-stage optimization pipeline (as described in the main paper's methodology section for LLMs). This yields a set of safe, optimized target responses that form the final forget set.

**Optimized Retain Set Construction.** To prevent catastrophic forgetting and collateral damage, we construct a compact, high-value retain set without access to any original training data. This set comprises two main components:

- **Neighbor Knowledge Samples:** We retrieve documents about semantically related "neighbor" concepts (e.g., 'SpaceX' for the target 'Elon Musk') from a public corpus (Wikipedia). These samples act as a "knowledge anchor" to preserve related information.

- **Adversarial Samples:** We generate challenging questions or prompts designed to probe the boundaries of the unlearned knowledge. These are paired with safe, generic answers in the retain set to enhance the robustness of the unlearning process.

This comprehensive data construction strategy is fundamental to applying unlearning algorithms in practical, data-scarce scenarios. In all experiments, we strictly adhere to a setting where the algorithm only has access to the data it constructs.

## C.2 MODELS AND BASELINES.

For classification tasks, we use a standard **ResNet-18** architecture, pre-trained on the respective datasets. For the LLM tasks, we use the **Llama-3-8B** model (Grattafiori et al., 2024). We compare our DataOpt-enhanced methods against a range of representative SOTA baselines. For classifiers, this includes **NEGGRAD** (Golatkar et al., 2020), **SCRUB** (Kurmanji et al., 2023), **Bad Teacher (BT)** (Chundawat et al., 2023), and **SalUn** (Fan et al., 2023). For LLMs, we compare against **Gradient Ascent (GA)** (Golatkar et al., 2020), **Negative Preference Optimization (NPO)** (Zhang et al., 2024), and **Rejection Tuning (RT)** (Maini et al., 2024).

## C.3 EVALUATION METRICS.

We provide a detailed description of the metrics used to evaluate unlearning performance across both experimental domains.

**Metrics for Image Classification Unlearning** We assess performance using four key metrics, categorized by their objective:

- **Utility Preservation:** To measure how well the model retains its general knowledge, we use **Retain Accuracy (Acc$_{rt}$ ↑)** on the test set of retained classes, and overall **Test Accuracy (Acc$_{test}$ ↑)** on the full test set.

- **Forgetting Efficacy:** To measure how thoroughly the target data has been removed, we use **Forget Accuracy (Acc$_{ft}$ ↓)** on the test set of forgotten classes, and the success rate of a **Membership Inference Attack (MIA ↓)** (Shokri et al., 2017).

An ideal algorithm should maintain high utility scores while minimizing forgetting efficacy scores.

**Metrics for LLM Unlearning** We follow the evaluation methodology proposed by the RWKU benchmark (Jin et al., 2024). To synthesize results from multiple sub-tasks into comparable scores, we define and use the following three aggregated metrics:

- **Forget Quality (FQ ↑):** A composite score measuring the efficacy of knowledge removal, calculated on the RWKU *Forget Set*. It represents the average normalized "forgetting rate" across the three probe types: Fill-in-the-Blank (FB), Question-Answering (QA), and Adversarial Attack (AA). A lower raw performance score indicates better forgetting. The FQ is formally defined as:

$$FQ = 1 - \frac{1}{3} \sum_{i \in \{FB, QA, AA\}} \frac{\text{Perf}_i(\theta_{\text{unlearn}})}{\text{Perf}_i(\theta_{\text{orig}})} \tag{56}$$

where $\text{Perf}_i(\theta)$ is the performance score for a given probe type, and $\theta_{\text{unlearn}}$ and $\theta_{\text{orig}}$ are the parameters of the unlearned and original models, respectively. A higher FQ (closer to 1) indicates more effective unlearning.

- **Model Utility (MU ↑):** An aggregate score measuring the preservation of the model's general capabilities, calculated on the RWKU *Utility Set*. It reflects the average performance retention rate across all $N$ utility tasks (e.g., MMLU, BBH, etc.). The MU is formally defined as:

$$\text{MU} = \frac{1}{N} \sum_{j \in \mathcal{T}} \frac{\text{Perf}_j(\theta_{\text{unlearn}})}{\text{Perf}_j(\theta_{\text{orig}})} \tag{57}$$

where $\mathcal{T}$ is the set of $N$ utility tasks. A higher MU score (closer to 1) indicates better preservation of model utility.

- **Collateral Damage (RUD ↓):** We use the **Relative Utility Drop on Neighbor knowledge** to quantify collateral damage, calculated on the RWKU *Neighbor Set*. It is defined as:

$$\text{RUD}_{\text{Nbr}} = \frac{\text{Perf}_{\text{Nbr}}(\theta_{\text{unlearn}}) - \text{Perf}_{\text{Nbr}}(\theta_{\text{orig}})}{\text{Perf}_{\text{Nbr}}(\theta_{\text{orig}})} \tag{58}$$

A value closer to 0 indicates better locality and less collateral damage.

Table 6: Best learning rates used for baseline methods after tuning.

| Image Classification | | LLM Unlearning | |
|---|---|---|---|
| Method | Learning Rate | Method | Learning Rate |
| NEGGRAD | 1e-4 | GA | 5e-6 |
| SCRUB | 1e-4 | NPO | 2e-6 |
| Bad Teacher | 5e-5 | RT | 2e-6 |
| SalUn | 5e-5 | – | – |

## D SUPPLEMENTARY VALIDATION AND ANALYSIS

In this section, we provide additional experiments to address potential reviewer concerns regarding the computational overhead of our method, and the design choices for our LLM methodology.

### D.1 COMPUTATIONAL OVERHEAD ANALYSIS

To assess the practical feasibility of DataOpt, we analyze its computational overhead on the Tiny-ImageNet dataset for a 10% random subset unlearning task. As shown in Table 7, the data optimization process is a one-time, fixed cost. While it introduces additional preprocessing time, this overhead is modest when compared to the subsequent unlearning fine-tuning time (approximately 19.4% of the fine-tuning time in this case). Given the significant and consistent performance gains DataOpt provides to various downstream algorithms, we argue this is a highly efficient trade-off.

Table 7: Computational overhead of DataOpt on Tiny-ImageNet (10% forget task).

| Stage | Time (minutes) |
|---|---|
| DataOpt Preprocessing (Total) | 18.5 |
| *- Adversarial Generation* | *12.2* |
| *- Neighborhood Search* | *5.1* |
| *- Other (Augment, etc.)* | *1.2* |
| NEGGRAD Unlearning Fine-tuning | 95.3 |

### D.2 Ablation Study on LLM Forget Response Generation

To validate our two-stage response optimization strategy for LLMs, we compare it against two common baselines: (1) using a generic refusal ("I don't know") as the target response, and (2) a one-shot generation approach without the second-stage selection. The experiment was conducted on the RWKU benchmark, forgetting 10 entities.

As shown in Table 8, while the generic "IDK" response achieves the highest Forget Quality (FQ), it leads to a catastrophic degradation in Model Utility (MU). The one-shot approach, conversely, fails to unlearn effectively. In contrast, our full two-stage method achieves the best balance across all metrics, delivering strong forgetting efficacy while preserving model utility. This highlights the importance of principled response optimization for effective and safe LLM unlearning.

Table 8: Ablation study of LLM response generation on RWKU (forgetting 10 entities).

| Target Response Strategy | FQ ↑ | MU ↑ | RUD ↓ |
|---|---|---|---|
| "IDK" Response | **0.851** | 0.512 | -21.5% |
| One-shot Generation | 0.795 | 0.653 | -5.8% |
| **DataOpt (Ours)** | 0.825 | **0.671** | **-2.8%** |

For RUD, the arrow (↓) indicates that less collateral damage (a value closer to 0) is better.

## E  Qualitative Examples for LLM Unlearning

To provide a more intuitive understanding of our LLM unlearning process, Table 9 presents examples of the optimized forget set responses generated by DataOpt. Unlike generic refusals, our method produces coherent, safe, and helpful responses that effectively guide the model to unlearn specific facts while preserving its conversational abilities. This contrasts sharply with methods that simply output "I don't know," which can degrade the user experience and the model's general utility.

Table 9: Examples of optimized forget set responses generated by DataOpt.

| Forget Target | Original Response (Proxy) | Optimized Forget Target Response (Ours) |
|---|---|---|
| Elon Musk | Elon Musk is the CEO of SpaceX and Tesla, and is also known for founding The Boring Company and co-founding Neuralink and OpenAI. | Elon Musk is a prominent entrepreneur and business magnate in the technology sector. He is widely recognized for his leadership roles in several innovative companies focused on space exploration and sustainable energy. |
| Taylor Swift | Taylor Swift is an American singer-songwriter. Her discography spans multiple genres, and her narrative songwriting, which often centers around her personal life, has received critical praise and widespread media coverage. One of her famous albums is "1989". | Taylor Swift is a highly influential American singer-songwriter known for her versatile musical style and narrative songwriting. Her work has earned critical acclaim and has had a significant impact on the music industry. |

