# OpenReview forum: "Data-Centric Unlearning: Optimizing Labels and Retain Data via Learning Dynamics"
_ICLR.cc/2026/Conference — ICLR 2026 Conference Withdrawn Submission_

### Official Review · Reviewer_XLa5 · 2025-10-21

**Soundness:** 3
**Presentation:** 3
**Contribution:** 3
**Rating:** 4
**Confidence:** 3

**Summary:**

The paper is well-motivated, turns learning-dynamics (LD/RLD) analysis into concrete, data-layer rules for unlearning, and shows consistent gains across multiple baselines and modalities (classifiers and an LLM setting). The idea is sound and practically usable as a drop-in data optimisation front end for existing unlearning methods. Main reservations are the reliance on one-step LD assumptions, heuristic aspects and engineering overhead in the LLM pipeline, and limited discussion of operational costs and failure modes.

**Strengths:**

- The paper offers a set of data-centric unlearning rules (e.g., retain-label smoothing, neighbor/boundary mining) derived from learning dynamics (RLD), with clear theoretical backing and an intuitive explanation of each mechanism.

- The proposed framework wraps around a variety of existing unlearning baselines and introduces a tunable knob to adjust the trade-off between forgetting and retaining.

- Experiments demonstrate consistent improvements across multiple baselines and datasets, enhancing both privacy/locality metrics and overall utility, with minimal performance degradation.

- The method is designed to be minimally invasive, requiring no changes to core model training pipelines, making it easily adoptable by practitioners and scalable to larger models.

**Weaknesses:**

- The paper builds directly on Ren & Sutherland (2025) and similar recent LLM dynamics literature. While the "rule-guided" aspect is new in framing, the practical recipe is largely a reweighting and scheduling heuristic over existing unlearning baselines. The novelty is seems modest.

- The main concern for me is that the analysis rests on single-sample, small-step SGD with a “stable relative influence” assumption, then drops O(η^2) terms. Modern training (mini-batches, Adam/AdamW, momentum, weight decay, label smoothing) alters the effective dynamics; the paper neither extends the theory to these settings nor shows that conclusions still hold.

- The two-stage “generate candidates then score” approach lacks the clearer optimality story present for classifiers and appears sensitive to prompt/scorer choices.


Minor:
-You seems mix 𝑓_𝜃 and 𝜋_𝜃. Pick one to ensure the consistency
-The dimensions of A,K,G  are not defined....
- The last term of (2) O(η2...) appears without assumptions (smoothness/Lipschitz of the Jacobian).

I am open to raising my score if the authors can convincingly address the concerns

**Questions:**

see weakness

---

### Official Review · Reviewer_oaia · 2025-10-29

**Soundness:** 2
**Presentation:** 3
**Contribution:** 2
**Rating:** 4
**Confidence:** 4

**Summary:**

The paper proposes a data-centric unlearning pipeline that jointly optimizes labels for forget/retain sets and curates retain data (neighbor/boundary/adversarial) guided by learning-dynamics heuristics. It reports modest improvements over several baselines on image classifiers and a heuristic extension to LLMs.

**Strengths:**

- The paper presents a coherent, data-centric perspective that jointly optimizes forget/retain labels and retain-set composition. The rules are simple and practical to implement.
- The empirical evaluation is comprehensive, including multiple datasets/baselines.

**Weaknesses:**

- The paper lacks a clear unlearning formulation aligned with standard goals (e.g., retrain-from-scratch indistinguishability). The proposed objective is heuristic and not shown to target the canonical criterion.

- The results in Table 1 are not strong. The gains are modest and no confidence intervals or multi-seed reporting, so it’s unclear if improvements are statistically meaningful.

- Theoretical scope and assumptions are not stress-tested. For example, boundary claims use linear/softmax analyses and extend to deep nets qualitatively. It would be helpful to add experiments that directly measure boundary proximity (e.g., margin estimates or confidence entropy) vs observed “retain-set protection” to substantiate the link.

- Some important details are under-specified or missing (e.g., the MIA protocol), which makes comparisons hard to interpret.

**Questions:**

1. Please justify Eq. (4) and explain how your optimization objective connects to the canonical unlearning target.

2. Which MIA did you use? Please describe the exact protocol to enable replication and fair comparison.

---

### Official Review · Reviewer_GT5c · 2025-11-01

**Soundness:** 3
**Presentation:** 2
**Contribution:** 2
**Rating:** 4
**Confidence:** 4

**Summary:**

This paper addresses a critical gap in machine unlearning—overlooking the quality of training data—by proposing a data-centric framework (DataOpt) grounded in learning dynamics theory. The proposed framework systematically optimizes two key components: label assignment for both forget and retain sets, and strategic selection of retain samples. DataOpt can be applied to both classifier and LLM unlearning scenarios. The experiments conducted on CIFAR - 10, Tiny - ImageNet, and the RWKU benchmark demonstrate that DataOpt outpereforms existing unlearning methods. Sensitivity analyses also confirm the efficacy and controllability of the framework.

**Strengths:**

1. The paper provides a theoretical justification for label and retain sample optimization based on learning dynamics, leading to closed-form solutions for optimal label assignments
2. The experimental section is wide-ranging, with experiments across standard vision benchmarks (CIFAR-10, Tiny-ImageNet) and challenging LLM benchmarks
3. The paper provides detailed appendices for mathematical proofs, algorithmic steps, experimental details, and implementation settings.

**Weaknesses:**

1. The design of New Response Generation and adversarial sample generation for LLMs lacks sufficient rationality.  The method of relying on a one-time generation via LLMs followed by scoring and ranking does not ensure that the generated content aligns with the intended optimization goals. There is no mechanism to enforce diversity among candidates or verify the thoroughness of sensitive information removal. This may lead to suboptimal or unsafe responses that retain hidden sensitive knowledge.
2. The writing of the Method section is disorganized, with several critical issues impairing clarity. Label assignment for forget set and retain set are conflated, key parameters (e.g., PGD for classification, response generation for LLMs) lack justification in the main text and are only mentioned in appendices. The writing makes it hard to follow how components integrate into the DataOpt framework.
3. For LLM unlearning tasks, the comparison against baseline algorithms is overly basic and omits recent state-of-the-art methods, such as [1-3], which prevents a comprehensive evaluation of whether the proposed method outperforms the latest alternatives. This gap limits the ability to validate the method’s superiority in real-world LLM unlearning scenarios.

Refs:

[1] A closer look at machine unlearning for large language models. In The Thirteenth International Conference on Learning Representations, 2025.

[2] LLM unlearning via loss adjustment with only forget data. In The Thirteenth International Conference on Learning Representations, 2025

[3]  The wmdp benchmark: measuring and reducing malicious use with unlearning. In Proceedings of the 41st International Conference on Machine Learning

**Questions:**

Please refer to Weaknesses for details.

---

### Official Review · Reviewer_JZnz · 2025-11-01

**Soundness:** 2
**Presentation:** 3
**Contribution:** 2
**Rating:** 4
**Confidence:** 2

**Summary:**

DataOpt reframes machine unlearning as a data-centric optimization: it first assigns degree-controlled target labels to forget samples via an “unlearning degree” k and then selects retain data that are near the forget samples and decision boundaries (including adversarial examples), which most strongly regulate unintended shifts. This yields controllable forgetting—higher k smoothly increases privacy (lower forget accuracy/MIA) with only minor utility loss—while consistently boosting existing unlearning methods in experiments.

**Strengths:**

The paper tackles a pivotal but underexplored facet of machine unlearning: how the training data for unlearning is constructed. This is crucial for both privacy-driven deletions and surgical updates to deployed models. By elevating data selection and labelling to first-class design variables, the work addresses a clear gap with tangible practical value. Notably, its retain-set strategy is boundary-focused and derived from learning-dynamics analysis rather than heuristics.

**Weaknesses:**

My primary concern is the “one-step influence stability” assumption. if the model moves only a tiny amount, first-order terms dominate and the analysis holds. However, in realistic training: multiple epochs, learning-rate schedules, adaptive optimisers (e.g., Adam), and data-order effects, the influence of a single update can shift substantially. What is true after one small step may not persist after thousands.
For LLMs, the two-stage approximate optimisation  feels heuristic relative to the cleaner classification treatment. Using the same (or a closely related) LLM to score fluency/relevance risks circularity and leakage, and the formal guarantees do not carry over to sequence models.
On the empirical side, the main results lack error bars or statistical tests; multiple seeds and explicit variance reporting would materially strengthen the claims.
Clarifying point: In Table 3, the rows for DataOpt (Retain set only) and DataOpt (Boundary sample only) appear identical line-for-line.

**Questions:**

See the weaknesses.

---

### Note · Authors · 2025-11-19

**Comment:**

We would like to sincerely thank all the reviews for the time and care theydevoted to reviewing our paper. After carefully consideration on all the  comments and suggestions, we have decided to withdraw our paper at this stage in order to further improve the work. The reviews are very valuable to us, and we will draw on it closely as we revise and extend this research.

**Withdrawal Confirmation:**

I have read and agree with the venue's withdrawal policy on behalf of myself and my co-authors.